# The Preparation of a Low-Cost, Structurally Simple Triboelectric Nanogenerator Based on Fullerene Carbon Soot-Doped Polydimethylsiloxane Composite Film

**DOI:** 10.3390/ma17112470

**Published:** 2024-05-21

**Authors:** Shujie Yang, Wen Zhao, Oleg Tolochko, Tatiana Larionova

**Affiliations:** 1Department of Physics and Materials Technology, Institute of Machinery, Materials and Transport, Peter the Great St. Petersburg Polytechnic University, St. Petersburg 195251, Russia; yangshujie216@gmail.com (S.Y.);; 2Department of Mechatronics Engineering, College of Mechanical and Electrical Engineering, Northeast Forestry University, No. 26 Hexing Road, Xiangfang District, Harbin 150040, China

**Keywords:** triboelectric nanogenerator (TENG), fullerene carbon soot (FS), polydimethylsiloxane (PDMS), nanomaterials

## Abstract

Triboelectric nanogenerators (TENGs) have emerged as viable micro power sources for an array of applications. Since their inception in 2012, TENGs have been the subject of significant advancements in terms of structural design and the development of friction materials. Despite these advancements, the complexity of their structural designs and the use of costly friction materials hinder their practical application. This study introduces a simplified TENG model utilizing an economical composite film of fullerene carbon soot (FS)-doped polydimethylsiloxane (PDMS) (FS-TENG). It confirms the FS-TENG’s ability to convert mechanical energy into electrical energy, as demonstrated through experimental validation. The generated electricity by the FS-TENG can power devices such as light-emitting diodes (LEDs), digital watches, kitchen timers, and sports stopwatches, highlighting its efficiency. This research enhances the development of TENGs featuring low-cost, streamlined structures for sustainable and autonomous energy sensing applications.

## 1. Introduction

The burgeoning expansion of the Internet of Things (IoT) and artificial intelligence (AI) technologies is exerting considerable pressure on the power supply systems of distributed electronic devices [1]. To mitigate the environmental pollution and high expenses associated with frequent battery replacements or routine recharging [2], it is imperative to address the challenges posed by unsustainable power sources. Mechanical energy harvesting emerges as a pivotal solution for achieving sustainable power supply [3]. Among various energy harvesting mechanisms, TENGs have garnered significant attention due to their versatile structural designs, straightforward manufacturing processes, low production costs, and the broad spectrum of materials that can be utilized.

TENGs, leveraging the synergistic effects of triboelectrification and electrostatic induction, represent a novel class of electronic devices capable of harvesting mechanical energies such as human motion, tire rotation, wind, tidal forces, vibrations, and ultrasonic waves [4,5,6]. These energies can be efficiently transformed into electricity or electrical signals [7], showcasing wide-ranging applications in sustainable energy and autonomous sensing domains [8]. In 2012, Prof. Zhonglin Wang first proposed the concept of “triboelectric nanogenerators (TENGs)”, and promoted nonenergy as “distributed mobile energy in the era of Internet of Things, sensor networks, and big data”, which opens up a new chapter of human energy model [9]; this technology has garnered significant interest within the nano-energy sector. Research efforts have encompassed the development of composite friction materials [10,11,12], the surface modification of these materials [13,14], and the introduction of innovative structural designs [15].

Recently, triboelectric nanogenerator (TENG) technology has advanced significantly in micro- and nano-scale distributed power supply, self-driven sensing systems, high-voltage power supply, and blue energy harvesting. This novel technology allows for the extraction of high-entropy micro-nano mechanical energy, serving as a self-reliant power source for numerous IoT wireless sensors. TENGs can convert mechanical energy into electrical signals directly, eliminating the need for additional signal conversion circuits by producing electrical outputs corresponding to the environmental mechanical forces. Utilizing this feature, researchers have developed TENG-driven sensing systems for real-time monitoring of respiratory diseases [16]. Moreover, Luo et al. have created a durable, high-performance medical device leveraging the high voltage and low current properties of TENG [17]. In the realm of large-scale energy collection, a TENG-based device has been engineered to harvest significant quantities of blue energy underwater [18].

Recent advances in materials science have enhanced the efficiency and reliability of TENGs, facilitating their integration into diverse applications including wearable devices, autonomous sensors, and energy harvesting systems. Concurrently, researchers are exploring synergies between TENGs and other energy harvesting technologies to develop hybrid systems with superior energy conversion capabilities. Efforts are underway to augment the performance and adaptability of TENGs across various practical uses. Nonetheless, challenges such as material selection, stability, durability, standardization, expandability, and cost of preparation remain focal points in ongoing research. Addressing these issues is pivotal to furthering the practical deployment of TENG technology.

In this study, we introduce the FS-TENG, characterized by its straightforward structural design, low manufacturing costs, ease of production, and high stability and durability. The FS-TENG excels in converting low-frequency mechanical energy into electrical energy, capable of powering multiple LEDs and suitable for use with electronic devices such as electronic watches, kitchen timers, and sports stopwatches. Additionally, the FS-TENG can produce varied electrical signals in response to changes in mechanical energy. Leveraging this capability, the FS-TENG serves effectively as a self-powered sensor for mechanical motion monitoring. We have fabricated TENGs using common materials and a simple design approach, demonstrating promising applications in areas of distributed power following a Wiener process and self-powered sensing.

## 2. Materials and Methods

### 2.1. Materials

In constructing the FS-TENG, Polytetrafluoroethylene (PTFE) was chosen as the friction material due to its properties, with dimensions of 150 µm thickness and 4 cm × 4 cm area, sourced from Ningxin New Material Technology Corporation, Huzhou, China. A 100 nm silver layer was deposited on one side of the PTFE via magnetron sputtering (VTC-600-2HD, sourced from Shenzhen Kejing Star Technology Company, Shenzhen, China) to function as an electrode. The opposite side served as the triboelectric layer. The counterpart electrode comprised a copper foil, provided by Jinqiao Copper Corporation, Wenzhou, China, of the same dimensions as the PTFE and a thickness of 100 µm. This foil was coated with a Polydimethylsiloxane (PDMS) composite film containing FS, applied through spin-coating. The PDMS, obtained from Dow Silicones Corporation, Midland, MI, USA, was formulated with a mass ratio of 9:1 between the main agent (SYLGARD™ 184 Silicone Elastomer Base) and the curing agent (SYLGARD™ 184 Silicone Elastomer Curing). The FS particles, approximately 40 nm in size, were synthesized by blending pure soot (Leader Nano Technology Company, Jinin, China) with 10% sodium dodecyl sulfate (Shanghai Jizhi Biochemical Technology Co., Ltd., Shanghai, China) for 10 h. The resulting PDMS + FS film samples are shown in Figure 1.

### 2.2. Fabrication of FS-TENG

The preparation of the FS-doped PDMS composite films is illustrated in Figure 2: Initially, the PDMS base and curing agents were measured in a mass ratio of 10:1. Subsequently, various mass fractions of FS—0.05%, 0.1%, 0.15%, 0.2%, 0.25%, and 0.3%—were individually added to the PDMS base. Following mechanical mixing, the FS-doped PDMS mixture underwent ultrasonic treatment for 1 h to ensure the homogeneous dispersion of FS within the PDMS base. The curing agent was then introduced to the mixture, which was allowed to degas for 30 min. Afterwards, 1.5 mL of the mixture was applied to one side of a copper foil via spin-coating. The coated copper foil was subsequently removed and placed in a vacuum drying oven at 80 °C for 4 h. During the experiment, the laboratory temperature was 25 °C and the air humidity was 20%.

The FS-TENG employs two composite laminates as substrates, integrating a pair of sponges positioned between the plates serving as elastomers. This configuration enhances the elastic contact and separation of the electrodes. Initially, the triboelectric layers are spaced 3 mm apart. The designed FS-TENG features a straightforward structure, facilitating easy construction and utilization of commonly available materials, thus supporting scalability in mass production.

A schematic of the TENG model is depicted in Figure 2. A custom-designed, mechanically rational vibrating device, fabricated using a 3D printer, ensures periodic contact and separation between the upper and lower electrodes. The microstructure of the FS nanoparticles was examined using scanning electron microscopy. PDMS surface morphology observed using atomic-force microscope (AFM-Nano surf Flex C3000, sourced from Nanosurf Technology Company, Liestal, Switzerland). The electrical output, including current and voltage generated by the TENG, was quantified with a multifunctional electrometer (UNI-T UT8000, sourced from UNI-T Technology Co., Ltd., Dongguan, China).

## 3. Results

### 3.1. Output Characteristics of FS-TENG

Upon applying a 2 N pressure at a frequency of 3 Hz to the FS-TENG using the vibration device, the Isc and Voc are illustrated in Figure 3. It is observed that the Isc value progressively increases with the addition of FS, achieving a maximum instantaneous peak of 2.2 µA at an FS concentration of 0.15%. Beyond this point, the peak value of Isc gradually declines with further increases in FS concentration, as depicted in Figure 3a. Similarly, Figure 3b shows that Voc follows the same trend as Isc, with the maximum instantaneous peak reaching 18.49 V at the same FS concentration of 0.15%.

The quantity of charge transferred between electrodes in a single power generation cycle of the TENG can be determined by integrating the Isc over time. Figure 3c illustrates the maximum charge transferred in a single TENG cycle across various mass fractions of FS. Initially, the charge transferred between the electrodes increases with the FS mass fraction, reaching a peak, and subsequently decreases as the FS mass fraction continues to rise. The optimum transferred charge of 1.18 nC is achieved at an FS mass fraction of 0.15%.

To evaluate the output power density of the FS-TENG, resistors were integrated into the external circuit, and the relationship between current and voltage across the circuit is depicted in Figure 4a. With the increment in the load resistance in the circuit, there is a significant increase in the voltage across the load, accompanied by a rapid decrease in current. The output power density attains its peak at 145 µW/m^2^ when the external circuit is connected to a load resistance of 63 MΩ, as shown in Figure 4b.

### 3.2. The Long-Term Stability and Durability of FS-TENG

To ensure the stability and durability of the designed FS-TENG in practical applications, we conducted a long-term test. The FS-TENG operated continuously for three hours at a frequency of 3 Hz, during which the two friction electrode materials experienced over 30,000 friction cycles. After three hours, the output electrical performance remained stable, confirming the FS-TENG’s high reliability. Furthermore, to assess potential changes in the surface morphology of the friction material after extensive use, we examined the friction material’s surface post-experiment using atomic force microscopy (AFM). Figure 5a,c illustrate the 3D surface morphology of the FS-PDMS and PTFE friction surfaces prior to testing, while Figure 5b,d depict their post-test 3D surface morphology.

Before and after the experiment, there were minimal changes in the surface morphology. The roughness of the PTFE surface, as recorded in Figure 5a, was initially 0.12 Ra and, following repeated friction, marginally decreased to 0.11 Ra as depicted in Figure 5b. Similarly, the initial roughness of the FS-PDMS material was 0.26 Ra, which slightly reduced to 0.25 Ra after the friction test. The FS-TENG, a designed friction electrode material, demonstrated a roughness of 0.25 Ra. The output performance of FS-TENG remained stable throughout extended experimental durations, with the material’s surface morphology exhibiting negligible wear. These observations confirm the excellent durability of FS-TENG, making it suitable for prolonged operational periods.

### 3.3. Working Principle of FS-TENG

The operating principle of the triboelectric nanogenerator relies on the combined effects of triboelectrification and electrostatic induction. Triboelectrification, a prevalent physical phenomenon in the natural environment, has been recognized for over two millennia, yet its underlying physical mechanisms remain subjects of ongoing debate [19,20]. Prof. Zhonglin Wang’s research has shown that electron transfer predominantly governs triboelectrification between solids [21]. Furthermore, Prof. Wang expanded the concept of displacement current, deriving from Maxwell’s equations—one of the ten foundational equations in physics. This work has established the theoretical underpinnings of friction nanogenerators and laid the groundwork for their further development [22].

The fundamental equation for Maxwell’s displacement current is presented as follows:(1)$$J_{D}=\varepsilon_{0}\frac{\partial E}{\partial t}+\frac{\partial P}{\partial t}$$

*P* represents the polarized electric field density, *E* denotes the electric field, and ε_0_ is the vacuum dielectric constant.

The concept of displacement currents facilitates the unification of electric and magnetic fields. It also reveals that in media with surface electrostatic charges, such as friction electric materials, displacement currents encompass the polarization density *Ps* attributable to surface electrostatic charges:(2)$$J_{D}=\varepsilon \frac{\partial E}{\partial t}+\frac{\partial Ps}{\partial t}$$

In Equation (2), ε is the dielectric constant of the material. The second term in the equation indicates that the polarized electric field induced by surface electrostatic charges can generate an electric current, providing a theoretical basis for the operation of friction nanogenerators.

The operating principle of the designed FS-TENG can be described as follows: Initially, the triboelectric layers and electrodes are in a charge equilibrium, with no potential difference between the electrodes, as depicted in Figure 6(i). Upon contact and exertion of an external force, triboelectrification leads to the accumulation of positive charges on the PDMS contact surface and negative charges on the PTFE contact surface, illustrated in Figure 6(ii). Given the stronger electronegativity of PTFE [23], upon physical contact between the two dielectric materials, positive charges transfer from the less electronegative material to the more electronegative one, resulting in dissimilar charge accumulation on their surfaces [24]. With equal amounts of oppositely charged surfaces nearly aligned, the potential difference at the interface remains zero, and no current flows in the external circuit.

Prof. Zhonglin Wang’s team investigated the atomic characteristic photon emission spectra during triboelectrification between two solids [25]. These spectra reveal that electrons are transferred from atoms in one material to another at the interface, enhancing our understanding of the surface charge transfer mechanism during contact between two dielectric materials.

Upon release of the external force, the two dielectric layers separate due to the restoring force. The electrostatic induction effect then causes charges of opposite sign to be induced on respective electrodes. Due to the induced potential difference, this drives electrons from the electrode with a lower potential to that with a higher potential via the external circuit, generating a transient current signal, as depicted in Figure 6(iii).

As the two dielectric layers are separated by their maximum distance, each layer induces an equal amount of anisotropic charge on the back electrode. Consequently, there is no potential difference between the two electrodes, resulting in no current generation in the external circuit, as depicted in Figure 6(iv). Upon recontact between the PDMS and PTFE layers, an opposing potential difference arises, prompting charges to move in the reverse direction, which generates a negative current, as illustrated in Figure 6(v). This cyclical process of contact and separation consistently produces an alternating current in the external circuit.

### 3.4. Simulation with COMSOL

The operational mechanism of the FS-TENG was simulated using COMSOL software (COMSOL Multiphysics 6.1), with the results depicted in Figure 7. The color scale represents a potential cloud map, illustrating the distribution of potential across the surface. Additionally, the lines in the figure indicate the electric field lines.

When the triboelectric layers are in complete contact, as depicted in Figure 7(i), the potential difference at the interface is zero because the frictionally excited equal dissimilar charges lie almost in the same plane. As the triboelectric layers begin to separate, with the electrostatic induction effect, an opposite number of charges will be induced on the respective electrodes, and these induced charges will excite an electric field between the electrodes, forming an induced potential difference, and electrons will flow from the electrode with a low potential to the electrode with a high potential through an external circuit, the potential difference between the electrodes gradually escalates. This potential difference reaches its equilibrium when the layers are separated back to their initial state, indicating that the voltage increases with the separation of the triboelectric layers, as illustrated in Figure 7(ii,iii). In contrast, Figure 7(iv) shows that the potential difference between the electrodes decreases when the triboelectric layers approach one another again, similarly to the process depicted in Figure 6(v).

### 3.5. FS-TENG Powers Microelectronic Devices

An FS-TENG with 0.15% FS was selected for the experiments. To convert the alternating current (AC) generated by the FS-TENG into direct current (DC), a rectifier bridge was integrated into the circuit. Additionally, a 10 µF capacitor was utilized to store the electricity produced. The circuit diagram illustrating the TENG operation is shown in Figure 8(i). For improved visibility of the FS-TENG operating the LEDs, it is recommended to turn off ambient lighting. The FS-TENG wires should be connected to the positive and negative terminals of the LEDs. When mechanical pressure is applied to the FS-TENG for 1350 s, the capacitor charges to 3 V, and upon activation of a switch, it can power 72 LEDs for 1 s. These LEDs (GNL-3012GD, green), whose operation voltage and power of each LED are 1.8 V and 30 mW, are connected partly in parallel partly in series. An optical photograph displaying the illuminated LEDs is provided in Figure 8(v). The LEDs operated without external power sources, powered exclusively by the electrical energy generated by the FS-TENG. This illustrates the FS-TENG’s ability to convert mechanical energy into electrical energy, sufficient to energize small-scale electronic devices.

To assess the potential applications of the FS-TENG beyond powering LEDs, we evaluated its ability to power various electronic devices. The batteries of a kitchen timer, sport timer stopwatch, and electronic watch were removed, and the FS-TENG’s leads were connected to the respective positive and negative terminals. Operating the TENG for 780 s allowed the capacitor to charge to 2 V. Upon activation of the circuit switch, the stored energy in the capacitor successfully powered an electronic watch for 6 s, as depicted in Figure 8(ii), a sport stopwatch for 5 s, as shown in Figure 8(iii), and a kitchen timer for 3 s, as indicated in Figure 8(iv). These results demonstrate the FS-TENG’s promising potential in powering electronic devices.

TENGs generate electricity without relying on fossil fuels and do not emit harmful substances, making them crucial for environmental protection and sustainable energy development. TENGs can harvest energy from various sources, including mechanical motion, wind, and wave energy, enabling operation under diverse environmental conditions.

To explore the electrical signal characteristics produced by the FS-TENG under various mechanical motions, an FS-TENG with a 0.15% FS concentration was selected for experimentation. A 10 µF capacitor was integrated into the external circuit, and its charging behavior at different frequencies is depicted in Figure 9. The results show that an increase in mechanical frequency leads to a quicker charging rate of the capacitor. Specifically, after 800 s of mechanical pressing, the FS-TENG operating at a frequency of 3 Hz generates almost double the electrical energy compared to that at 1 Hz. The FS-TENG yields distinct electrical signals in response to different mechanical pressure frequencies. Based on this capability, FS-TENG can generate an electrical signal that matches the response of mechanical energy. The amplitude and frequency of the electrical signal reflect the characteristics of the mechanical energy and can directly convert the triggering of the mechanical energy into an electrical signal, which is characteristic of a self-driven sensing system. The designed FS-TENG can also be utilized as a sensor to determine the movement frequencies of dynamic objects, such as automobile tires and sports shoes.

## 4. Discussion

In the present study, we developed a cost-effective, structurally straightforward triboelectric nanogenerator (TENG) using a composite film of polydimethylsiloxane (PDMS) doped with fullerene carbon soot nanoparticles. Different mass fractions of fullerene carbon soot were incorporated into the PDMS matrix. Optimal performance of the fabricated FS-TENG was observed with a fullerene addition of 0.15%. Furthermore, COMSOL Multiphysics^®^ simulation software, alongside theoretical analyses, was utilized to elucidate the working mechanism of the FS-TENG and to examine the surface charge transfer mechanism following the contact of dielectric materials. The designed FS-TENG demonstrated capability to power electronic devices such as LEDs, electronic watches, kitchen timers, and sports stopwatches.

This study shares similarities with prior research. Fu developed a biobased fingernail TENG glove capable of harvesting and storing mechanical energy from hand movements to illuminate an LED [26]. Smitha created a TENG from graphite-coated paper, achieving a maximum short-circuit current density of approximately 0.8 μA cm^−2^, which is adequate for charging capacitors and powering LEDs [27]. Viyada enhanced the power output of a TENG by incorporating oxide graphene into the PDMS matrix, achieving peak output voltage and current of 330 V and 8 μA/cm^2^ respectively, sufficient to power 90 green LEDs [28].

In this study, the optimal performance of the designed FS-TENG was observed at an FS addition of 0.15%. It is hypothesized that the superior performance can be attributed to the high dielectric constant of FS nanoparticles. Their integration into PDMS likely enhances the dielectric constant of the composite film, which in turn improves its capacitive properties.

To determine the distribution of FS particles within the PDMS matrix, the prepared PDMS composite films were examined using an optical microscope. Optical photographs of the FS-TENG composite films are presented in Figure 10. Panels (a), (b), and (c) of Figure 10 display the PDMS matrices doped with 0.05%, 0.15%, and 0.3% FS particles, respectively. As the concentration of FS nanoparticles increases, their distribution within the PDMS matrix becomes more pronounced and the spacing between FS nanoparticles decreases. At an inclusion level of 0.3% FS particles, significant agglomeration of FS particles is observed within the PDMS matrix.

The modeled distribution of FS particles within the PDMS matrix was mapped based on the observed distribution illustrated in Figure 10. As shown in Figure 11a, the spherical structure of FS nanoparticles forms a porous network in the PDMS, providing an extensive surface area that acts as charge trapping sites. These sites facilitate the generation of increased triboelectric charges during the triboelectrification process, consequently enhancing the energy conversion efficiency of the TENG [13,28,29]. However, beyond a 0.15% addition of FS nanoparticles, the inter-particle spacing diminishes (as illustrated with d2 < d1 in Figure 11), leading to electron migration between FS particles. This migration results in the formation of a conductive network, as shown in Figure 11b, inducing significant dielectric loss and adversely impacting the output performance of the FS-TENG. This observation is corroborated by the trend of the transferred charge between electrodes, which initially rises and then falls with increasing FS content, as demonstrated in Figure 3c.

Admittedly, the major limitation of the present study is the relatively low voltage of the TENG excitation, which registers significantly below the 100 V benchmark observed in prior research. The output power of the FS-TENG requires enhancement as it currently provides energy to small electronic devices, such as electronic watches and sports timers, for only 3–5 s. This performance must be improved to broaden its utility in the realm of micro and nano distributed energy systems. Notwithstanding these limitations, the designed FS-TENG demonstrates a good ability to convert mechanical energy into electrical energy.

The surface of the dielectric material remained unprocessed; enhancing this surface based on existing studies could potentially improve the roughness and augment the triboelectrification effect, thereby increasing the output performance of the triboelectric nanogenerator (TENG). This work demonstrates the capability of TENGs to power light-emitting diodes (LEDs) and electronic devices, such as electronic watches, kitchen timers, and sports stopwatches, contributing significantly to the advancement of TENG technology. Such achievements may encourage further research to optimize design, boost efficiency, and explore broader applications and scenarios.

## 5. Conclusions

In this study, cost-effective FS nanoparticles were utilized to enhance a TENG, which, despite its simplistic design, effectively converts mechanical energy into electrical energy. Experiments on the preparation of fullerene carbon soot-doped PDMS composite films and the output performance of the FS-TENG were conducted. The main conclusions are as follows:(1)The optimal FS-TENG performance is achieved when FS is added at 0.15%.(2)The developed FS-TENG exhibits a maximum Voc of 18.49 V and Isc of 2.2 µA, achieving a peak power density of 145 µW/m^2^ with a 63 MΩ load resistor.(3)The electrical energy generated by the FS-TENG is sufficient to illuminate 72 LEDs and can also power devices such as a digital watch, kitchen timer, or sports stopwatch.(4)Furthermore, the FS-TENG is capable of producing varying electrical signals in response to different mechanical energies harnessed, making it suitable as a sensor for monitoring mechanical movements.(5)This research contributes to the advancement of deploying TENGs with both low-cost and straightforward designs in sustainable energy and autonomous energy sensing applications.

## Figures and Tables

**Figure 1 materials-17-02470-f001:**
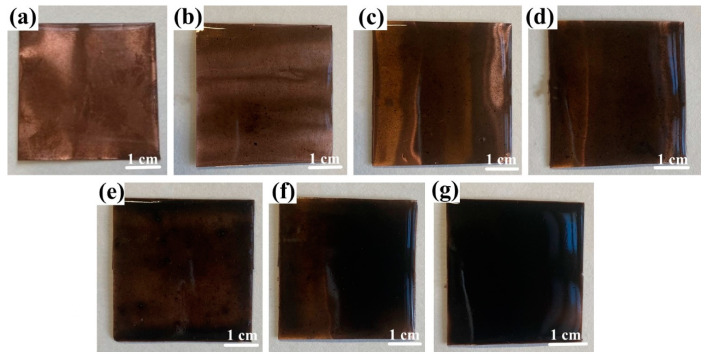
Optical photographs of PDMS+ FS films: (**a**) PDMS; (**b**) PDMS + 0.05% FS; (**c**) PDMS + 0.1% FS; (**d**) PDMS + 0.15% FS; (**e**) PDMS + 0.2% FS; (**f**) PDMS + 0.25% FS; (**g**) PDMS + 0.3% FS.

**Figure 2 materials-17-02470-f002:**
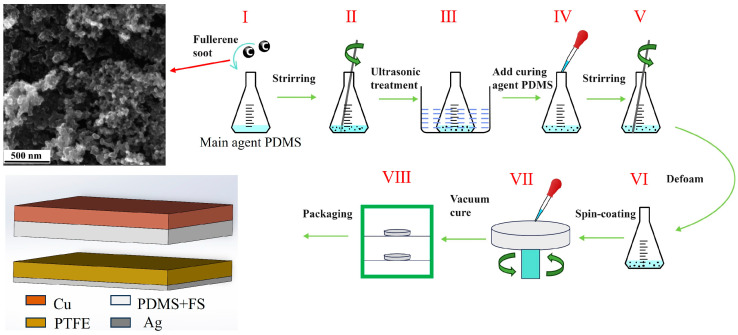
Schematic flow of FS-doped PDMS composite film preparation.

**Figure 3 materials-17-02470-f003:**
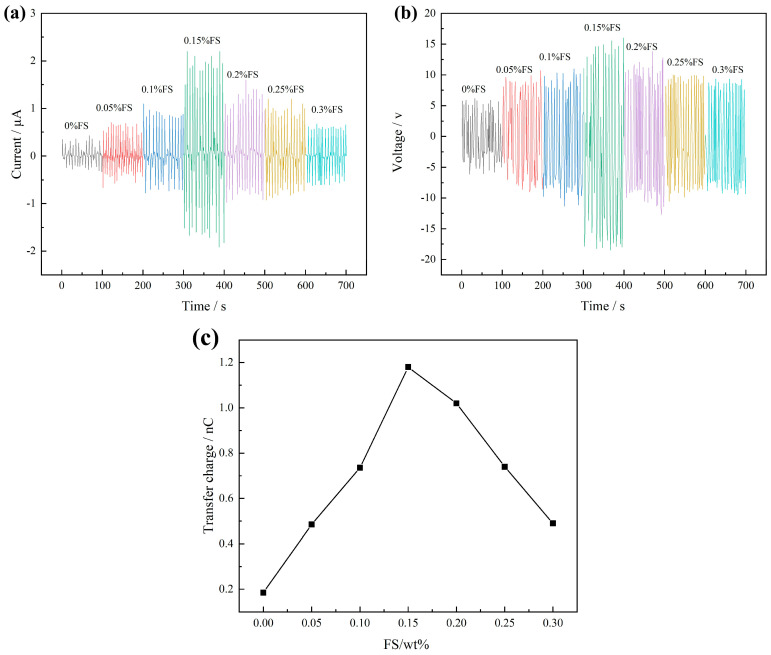
(**a**) Short-circuit current Isc; (**b**) open-circuit voltage Voc; (**c**) maximum transferred charges between electrodes.

**Figure 4 materials-17-02470-f004:**
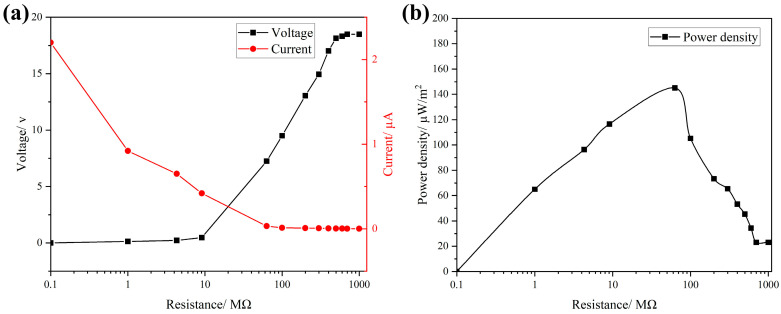
(**a**) Maximum Isc vs. Voc with different loads; (**b**) FS-TENG output power density.

**Figure 5 materials-17-02470-f005:**
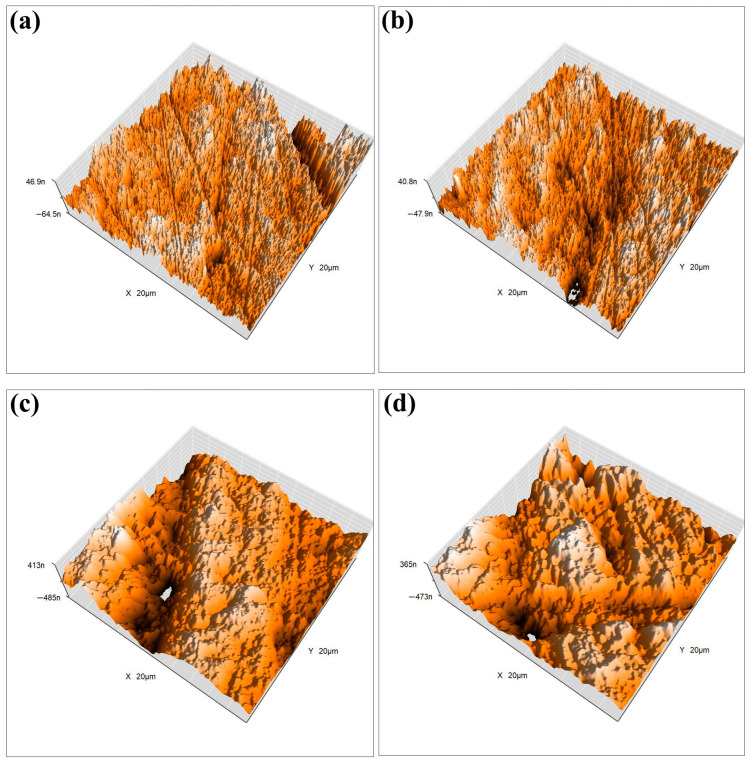
(**a**) The 3D surface morphology of FS-PDMS friction surfaces prior to experimentation; (**b**) 3D surface morphology of FS-PDMS friction surfaces following experimentation; (**c**) 3D surface morphology of PTFE friction surfaces prior to experimentation; (**d**) 3D surface morphology of PTFE friction surfaces following experimentation.

**Figure 6 materials-17-02470-f006:**
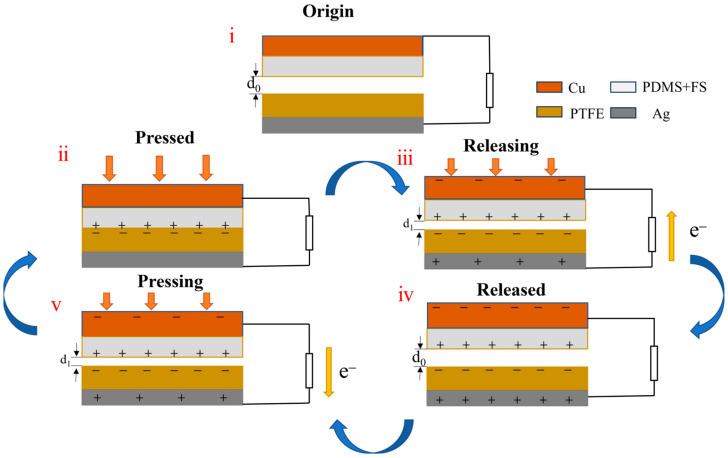
Schematic diagram of the FS-TENG working mechanism. (i) Triboelectric materials are in initial state; (ii) triboelectric materials are in full contact under external pressure; (iii) triboelectric materials are in separation when the force is released; (iv) triboelectric materials completely separate and return to the initial state; (v) triboelectric materials are brought close together again by an external force.

**Figure 7 materials-17-02470-f007:**
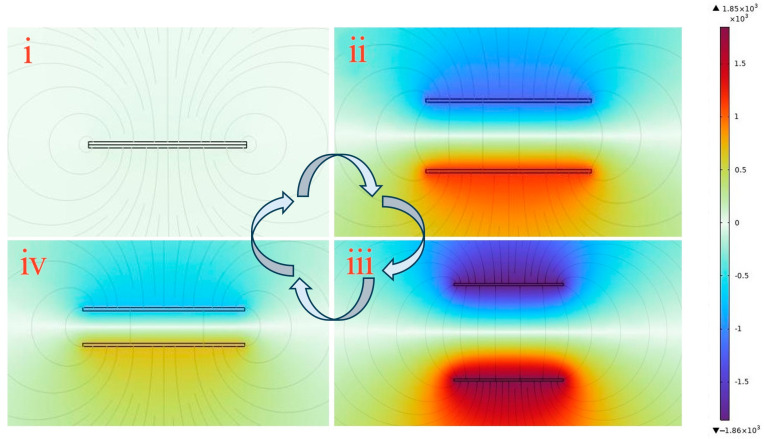
COMSOL simulation of FS-TENG working mechanism. (i) Triboelectric materials are in full contact under external pressure; (ii) triboelectric materials are in separation when the force is released; (iii) triboelectric materials completely separate and return to the initial state; (iv) triboelectric materials are brought close together again by an external force.

**Figure 8 materials-17-02470-f008:**
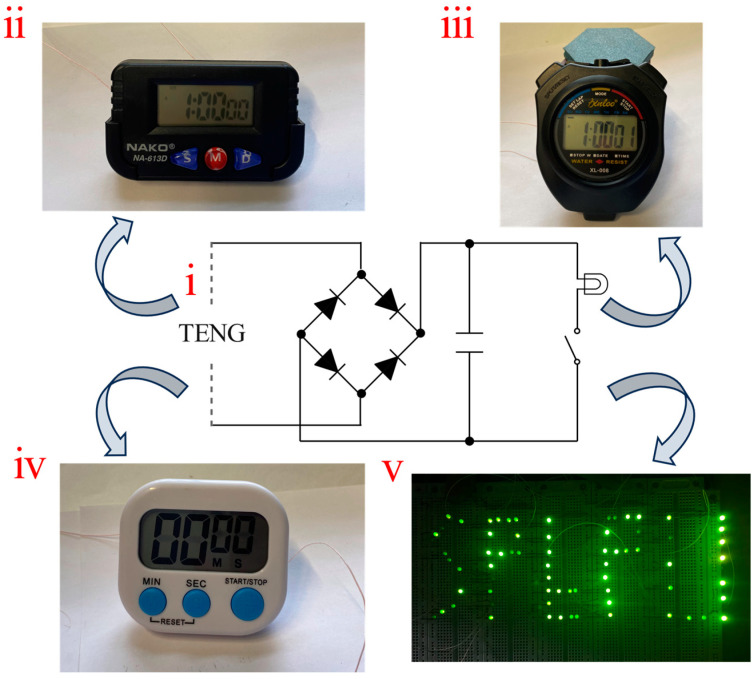
(i) Schematic diagram of the FS-TENG operating circuit; (ii) FS-TENG powers the electric watch; (iii) FS-TENG powers the sport stopwatch; (iv) FS-TENG powers the kitchen timer; (v) FS-TENG lights up the LEDs.

**Figure 9 materials-17-02470-f009:**
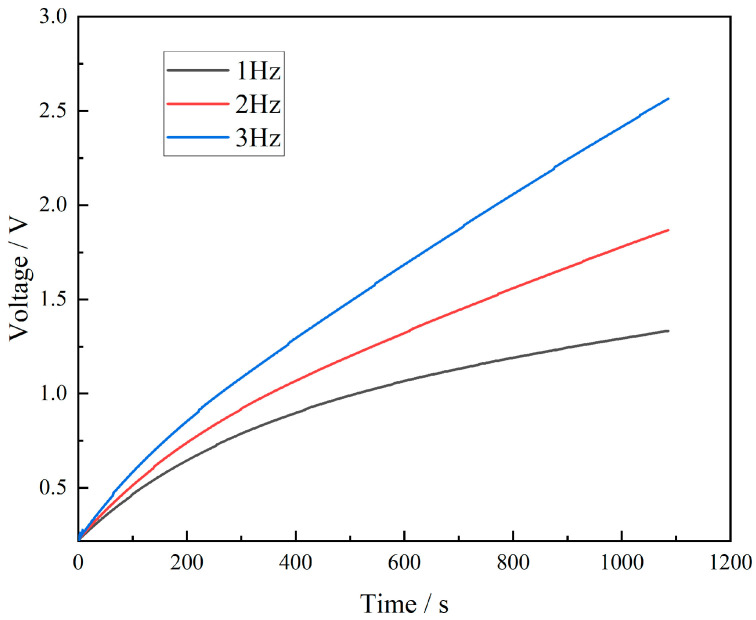
Capacitor charging curve.

**Figure 10 materials-17-02470-f010:**
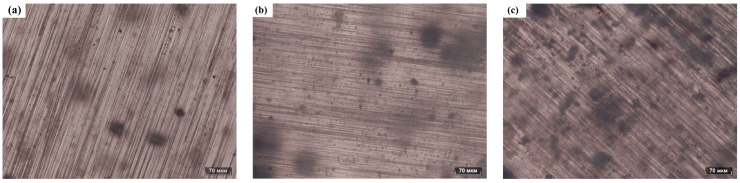
Optical photograph of FS-PDMS. (**a**) PDMS-0.05% FS, (**b**) PDMS-0.15% FS, (**c**) PDMS-0.3% FS.

**Figure 11 materials-17-02470-f011:**
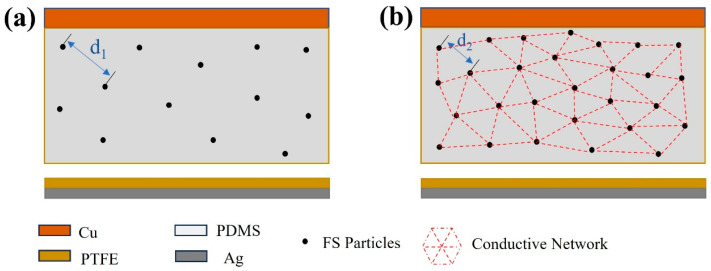
Theoretical modeling of the effect of FS on FS-TENG performance: (**a**) 0.15% FS; (**b**) >0.15% FS.

## Data Availability

Data are contained within the article.

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
