# Peer review of "The Preparation of a Low-Cost, Structurally Simple Triboelectric Nanogenerator Based on Fullerene Carbon Soot-Doped Polydimethylsiloxane Composite Film"

_materials, 2024, doi:10.3390/ma17112470_

Round 1

Reviewer 1 Report

Comments and Suggestions for Authors

Minor revision is required.

1.      Please consider the following title: “The preparation of a low-cost, structurally simple triboelectric nanogenerator based on fullerene carbon soot-doped polydimethylsiloxane composite film.”

2.      The authors should clarify the novelty of this current manuscript in the Introduction section.

3.      Caption of Figure 3C. It lacks “/” for the percentage. The caption should be FS/wt%.

4.      Figures 3c, 4a, 4b, and 8- the line should be thicker (use line 1.5 or 2).

5.      An English revision of the text should be done.

Comments on the Quality of English Language

The English language should be improved

Reviewer 2 Report

Comments and Suggestions for Authors

The authors presented a work titled "A low-cost structurally simple triboelectric nanogenerator based on fullerene carbon soot doped polydimethylsiloxane composite film" is a good effort. Here are some major and minor comments for the paper.

Major Comments:

The working principle and operational mechanism of the FS-TENG should be explained in more detail, supported by theoretical analyses and simulations.

The characterization of the FS nanoparticles and their dispersion in the PDMS matrix is lacking. Detailed morphological and structural analyses using techniques like TEM, XRD, and Raman spectroscopy would be beneficial.

The choice of PTFE as the triboelectric material should be justified, and a comparison with other commonly used materials should be provided.

The influence of FS concentration on the dielectric properties and triboelectric charge density of the PDMS composite should be investigated systematically.

The output performance of the FS-TENG should be compared with other state-of-the-art TENGs reported in the literature to highlight its advantages and limitations.

The long-term stability and durability of the FS-TENG under continuous operation should be evaluated, as these are crucial factors for practical applications.

The scalability of the FS-TENG fabrication process and the potential for large-scale production should be discussed.

The authors should explore and discuss potential applications of the FS-TENG beyond powering LEDs, such as in wearable electronics, environmental sensing, or IoT devices.

Minor Comments:

The introduction section should provide a more comprehensive review of the current state-of-the-art in TENGs, highlighting the gaps and challenges that the present work aims to address.

The experimental section should include more details on the materials used, their sources, and the specific experimental conditions.

The figures and captions could be improved for better clarity and readability.

The discussion section should be expanded to provide more insights into the observed results and the underlying mechanisms.

The conclusion section should be more concise and focused on the key contributions 

Reviewer 3 Report

Comments and Suggestions for Authors

I proceeded to analyze the manuscript entitled:

A low-cost structurally simple triboelectric nanogenerator based on fullerene carbon soot doped polydimethylsiloxane composite film,

written by: Shujie Yang, Wen Zhao, Oleg Tolochko and Tatiana Larionova

The manuscript deals with a fullerene soot (FS) doped polydimethylsiloxane (PDMS) (FS-TENG) generator. The authors claim that this generator, with a straightforward design, low production costs, and ease of fabrication. can converting low-frequency mechanical energy into electrical power and is capable of illuminating several LEDs. The authors also claim that FS-TENG can generate varying electrical signals in response to changes in the mechanical energy harvested, therefore it can be used as a self-powered sensor for monitoring mechanical movements.

The topic is, in my opinion, interesting. The figures are suggestive and support the statements. References are in proper amount and indicate that the authors are well aware of what has been published on the subject they are writing about. The article is well written, using good English, in my opinion, but a careful check would improve it. The content of the article sustains the Conclusion.

Moving to details, I found a few parts that, in my opinion, require improvement and additional clarification, as indicated on each item, and they are mentioned below.

-Abstract: “It demonstrates the capability of FS-TENG to convert mechanical energy into electrical energy via experimental validation. The FS-TENG achieves a maximum instantaneous voltage (Voc) of 18.49 V and current (Isc) of 2.2 μA, with a peak power density of 145 μW/m2 under a load resistance of 63 MΩ. The electricity produced by the FS-TENG can power 72 LEDs, underscoring its efficiency.”

This is a very detailed result. I believe that the abstract should be more general, explaining what the work is about, and such phrases should be located in the results section.

-Line 59: give details on how you produced FS.

-Line 67:”curing agents”. Give details on them.

-Line 138, Fig. 6, explain the color scale. Check the upper value on the color scale. Explain what the lines represent? Are they the electric field lines?

-Line 187, line 146: you mentione repteatedly that:”The electrical energy generated by the FS-TENG is sufficient to illuminate 72 LEDs” You presetned a photo, but write in mnuscript how were they connected? What type pf LEDs did you use, what is the operation voltage and power of each LED? If they were #V LEDs, most probably you connected them in parallel. How long did they illuminate for? Most likely you did a dark room video recording and extracted the frame where the LEDs did a short flash on the capacitor discharging. Clearly explain this in manuscript.

-Lines :171-173: “These sites facilitate the generation of increased triboelectric charges during the triboelectrification process, consequently enhancing the energy conversion efficiency of the TENG”

Write a short explanation on the mechanism of charge transfer between FS-PDMS and PTFE, qouting the papers you take it from. This will complete your manuscript making it more interesting for readers interested in the Material Science Physics.

Round 2

Reviewer 2 Report

Comments and Suggestions for Authors

The authors have addressed all the comments. I think the paper is now good for acceptance.

Reviewer 3 Report

Comments and Suggestions for Authors

Comment to the authors

I proceeded to re-analyze the manuscript entitled:

A low-cost structurally simple triboelectric nanogenerator based on fullerene carbon soot doped polydimethylsiloxane composite film,

written by: Shujie Yang, Wen Zhao, Oleg Tolochko and Tatiana Larionova

The authors answered the issues I raised and added clarification in manuscript.

I have no further comment.